# Platelet Activation and Cytokine Release of Interleukin-8 and Interferon-Gamma-Induced Protein 10 after ChAdOx1 nCoV-19 Coronavirus Vaccine Injection

**DOI:** 10.3390/vaccines11020456

**Published:** 2023-02-16

**Authors:** Chih-Lung Shen, Tso-Fu Wang, Chao-Zong Liu, Yi-Feng Wu

**Affiliations:** 1Department of Hematology and Oncology, Hualien Tzu Chi Hospital, Buddhist Tzu Chi Medical Foundation, Hualien 970, Taiwan; winpower0105@gmail.com (C.-L.S.); tfwang@tzuchi.com.tw (T.-F.W.); 2College of Medicine, Tzu Chi University, Hualien 970, Taiwan; 3Department of Pharmacology, School of Medicine, Tzu Chi University, Hualien 970, Taiwan; czliu33@gmail.com; 4Ph.D. Program in Pharmacology and Toxicology, Tzu Chi University, Hualien 970, Taiwan

**Keywords:** nCoV-19 coronavirus, vaccine, ChAdOx1, platelet activation, platelet factor 4, cytokine release

## Abstract

Coronavirus disease 2019 (COVID-19) vaccines are associated with serious thromboembolic or thrombocytopenic events including vaccine-induced immune thrombocytopenia and thrombosis and immune thrombocytopenia, particularly AZD1222/ChAdOx1. According to the proposed mechanism, COVID-19 vaccines stimulate inflammation and platelet activation. In this study, we analyzed the role of AZD1222/ChAdOx1 vaccines in the activation of platelets and the release of anti-PF4 antibodies and inflammatory cytokines in a cohort of healthy donors without vaccine-induced immune thrombotic thrombocytopenia (VITT). Forty-eight healthy volunteers were enrolled in this study. Blood samples were collected from peripheral blood at three time points: before vaccination and 1 and 7 days after vaccination. Compared with the prevaccination data, a decrease in the leukocyte and platelet counts was observed 1 day after vaccination, which recovered 7 days after injection. The percentage of activated GPIIb/IIIa complex (PAC-1) under high ADP or thrombin receptor-activating peptide stimulation increased 1 day after vaccination. Furthermore, interleukin-8 (IL-8) and interferon-gamma-induced protein 10 (IP-10) increased significantly. Additionally, platelet activation and inflammation, with the release of cytokines, were observed; however, none of the individuals developed VITT. Mild thrombocytopenia with platelet activation and inflammation with an elevation of IL-8 and IP-10 were observed after AZ vaccination.

## 1. Introduction

Severe acute respiratory syndrome coronavirus 2 (SARS-CoV-2) infection, which is associated with coronavirus disease-2019 (COVID-19), was first reported at the end of 2019. The disease spread rapidly across the world in a few months, thereby posing a major health threat to the global population [1]. Tracing the Weekly COVID-19 Epidemiological Update until 8 November 2022 revealed that over 630 million individuals were confirmed to have the disease and that more than 6.6 million associated deaths were reported from all six World Health Organization (WHO) regions [2,3,4,5]. Most patients with COVID-19 were classified as mild (81%), 14% as severe, and 5% as critical. The overall case fatality rate was 1.05%, and all deaths were reported in critically ill patients, which accounted for almost half of the critical cases [6]. The chimp adenovirus vector-based Oxford/AstraZeneca [AZD1222/ChAdOx1] (AZ) COVID-19 vaccine was approved by the United Kingdom in December 2020 and by the European Medicines Agency at the end of January 2021, and it is one of the most widely used COVID-19 vaccines.

However, the vaccines have been reported to be associated with serious thromboembolic or thrombocytopenic events including vaccine-induced immune thrombocytopenia and thrombosis (VITT) and immune thrombocytopenia (ITP). The first cases of severe thromboembolic events combined with thrombocytopenia following AZ vaccine administration were reported in March 2021 [7,8,9]. These events led to a pause in the use of the AZ vaccine in several European countries, and Denmark later from the general vaccination program. Subsequently, extensive case series on the occurrence and severity of this complication, now called VITT, were published; furthermore, vaccine-related ITP was reported. These side effects raise concerns of an increased risk of ITP after ChAdOx1 vaccination [10,11]. The mechanisms and pathogenesis of VITT are yet to be completely elucidated. Some studies have suggested that these events are related to platelet consumption and the progressive thrombogenic state involving the formation of antibodies against protein platelet factor 4 (PF4) [12]. PF4 is a tetrameric chemokine (CXCL4) stored mainly in megakaryocytes and alpha granules of platelets and released upon platelet activation [13]. In VITT, immunoglobulin G (IgG) antibodies against PF4–polyanion complexes are released, and platelets are activated via the surface Fcg receptor. Later, an increase in PF4 levels due to platelet activation occurs, which contributes to the formation of complexes with newly formed PF4 autoantibodies. These mechanisms have been speculated to cause thrombocytopenia, platelet aggregation, and thrombus formation [14].

According to this proposed mechanism, COVID-19 vaccines may be involved in inflammation and platelet activation. In this study, we analyzed platelet activation, aggregation markers, whole blood coagulation, anti-PF4 Abs, and cytokines released because of inflammation and compared them before and after the administration of AZD1222/ChAdOx1 vaccines in a cohort of healthy donors without VITT.

## 2. Materials and Methods

### 2.1. Healthy Volunteers

Between 1 June 2021, and 31 December 2021, healthy volunteers at the Hualien Tzu Chi Hospital, Taiwan were recruited for this study. Informed consent was obtained after explaining the study protocol and before the injection of the ChAdOx1 nCoV-19 coronavirus vaccine. None of the volunteers received other kinds of COVID-19 vaccines including mRNA vaccines. This study was approved by the Institutional Review Board of Tzu Chi General Hospital (IRB110-135-B). The samples were collected from peripheral blood at three time points: before vaccination and 1 and 7 days after vaccination. Complete blood count, differential leukocyte count, and D-dimer level were analyzed using Sysmex XN-9000. Blood samples were also sent for the below-mentioned study. Symptoms after vaccination were further recorded.

### 2.2. Antiplatelet Factor IV Antibody (Anti-PF4 Ab)

Anti-PF4 Ab was tested using enzyme-linked immunosorbent assays. The LIFECODES PF4 IgG assay (Immucor, HAT45G^®®^, Milan, Italy) showed a normal range (median ± standard deviation) of anti-PF4 Abs for the healthy controls and a range for patients receiving heparin. Plasma samples were obtained from the volunteers and tested according to the manufacturer’s instructions. The ODES PF4 IgG assay (Immucor, HAT45G^®®^, Milan, Italy) exhibited a normal range (median ± standard deviation) of anti-PF4 Abs. Optical density (OD) values < 0.35 were considered negative, and OD values ≥ 0.35 indicated significant levels of antibodies against PF4/PVS [12,15].

### 2.3. Platelet Surface Markers of Resting and Activated Platelets

The surface markers of resting platelets were evaluated using whole blood flow cytometry incubated with CD42a (Glycoprotein IX; GPIX), CD42b (GPIbα), and CD41 (GPIIb; IIb integrin). As previously described, platelet activation was also evaluated using whole blood flow cytometry [16]. Briefly, citrate-anticoagulated whole blood was incubated with an antibody cocktail that contained fluorescein isothiocyanate (FITC)-conjugated PAC1, phycoerythrin (PE)-conjugated anti-*p*-selectin, and PE-Cy5-conjugated anti-CD42b with or without 0.5 μm or 20 μm ADP or 1.5 or 20 μm thrombin receptor-activating peptide (TRAP) for 15 min at room temperature. The samples were fixed with 1% formaldehyde. All of the above antibodies were purchased from BD Pharmingen™ (Franklin Lakes, NJ, USA).

### 2.4. Platelet–Leukocyte Aggregation

The gating strategy has been shown to identify platelet aggregates with different leukocyte subtypes of interest. CD33 was used for determining the myeloid series including neutrophils, and CD14 was used for identifying monocytes. Furthermore, CD19 was used to identify B lymphocytes and CD3 to identify T lymphocytes to evaluate the lymphocyte–platelet aggregation.

Subsequently, 100-µL blood samples were incubated with an antibody cocktail including FITC-conjugated anti-CD14, PE-conjugated anti-CD33, PerCP-conjugated anti-CD45, and APC-conjugated anti-CD42b, at room temperature. The samples were lysed with FACS Lysing Solution (Becton Dickinson, BD Biosciences, Oxford, UK). After centrifugation for 5 min at 400× *g*, the platelet pellet was resuspended in 500-µL HEPES-buffered saline solution and then analyzed using flow cytometry. All of the above antibodies were purchased from BD Pharmingen™.

### 2.5. Flow Cytometry

Flow cytometric analyses of the platelet surface antigen were performed. The samples were evaluated using a FACSCalibur flow cytometer (Becton Dickinson Immunocytometry Systems, San Jose, CA, USA). Data were analyzed using the WinList platform (Verity Software, Topsham, ME, USA).

### 2.6. Cytokine Protein Levels

Merck Milliplex Human Cytokine/Chemokine Panel (HCYTA-60K-10) was evaluated according to the manufacturer’s instructions. All validation-related measurements were performed using the magnetic bead-based kit and run on a MAGPIX^®®^ (Austin, TX, USA) Instrument. Cytokine antibodies including IFNr, interleukin (IL)-1b, IL-6, IL-8, IL-15, interferon-gamma-inducible protein 10 (IP-10), macrophage inflammatory protein-1 (MCP-1), MIP-1b, RANTES, and tumor necrosis factor-alpha (TNF-α) were conjugated on the premixed beads for this assay kit.

The samples were analyzed using a Luminex 200 instrument (Luminex Corporation, Austin, TX, USA). Each sample was run as a single measurement, and the median fluorescence intensity (MFI) data were analyzed using a five-parameter logistic curve-fitting method. The individual corresponding cytokine concentrations in the blood samples were calculated from the MFI data.

### 2.7. Statistical Analysis

The collected data were entered into GraphPad Prism, version 8, for analysis. Laboratory data, platelet surface antigens, platelet–leukocyte aggregations, and cytokines between the different time points were compared using one-way ANOVA. Data were expressed as the means and 95% confidence intervals (95% CIs). *p*-values of <0.05 were used to denote statistical significance.

## 3. Results

### 3.1. Clinical Data

Forty-eight healthy volunteers were enrolled in this study. The baseline characteristics of the volunteers included 13 men and 35 women. Their mean age was 39.1 ± 9.8 years. The side effects included 50.0% local reddish and pain over injection sites, 33.3% fatigue, 12.5% headache, 25.0% muscle pain, 16.7% chills, 12.5% joint pain, and 8.30% fever. The above minor side effects were noted in 72.9% (*N* = 35) of the volunteers. No severe side effects including severe allergy, drowsiness, abdominal pain, poor appetite, lymphadenopathy, or VITT were found. In all 48 donors, peripheral blood samples were collected at three time points: before vaccination and 1 and 7 days after the administration of the ChAdOx1 nCoV-19 coronavirus vaccine. Compared with the prevaccination data, the leukocyte and platelet counts decreased 1 day after vaccination (leukocyte count: 6946 ± 1887 vs. 6333 ± 1748; *p* = 0.007, and platelet count: 285,000 ± 65,000 vs. 276,000 ± 66,000; *p* = 0.0182) and recovered 7 days after vaccination. A mild decrease in the mean cell volume was observed 7 days after vaccination (12.24 ± 1.23) compared with the prevaccination data (9.92 ± 1.03) (*p* = 0.002). Differential leukocyte counts showed that only the percentage of monocytes increased 1 day after vaccination (Detailed data are presented in Table 1.)

Furthermore, only one sample had increased D-dimer levels (>2000 mg/L FEU) 1 day after vaccination. The cut-off point for the D-dimer was selected according to the VITT guidelines [17]. However, no thrombotic or bleeding events were noted in any of the healthy volunteers. The sample with a high D-dimer level was from the seventh volunteer (Figure 1A). Tracing this volunteer, the platelet counts were 206,000, 193,000, and 223,000/μL at the above three time points, respectively.

### 3.2. Anti-PF4 Ab

The data pertaining to anti-PF4 Ab showed that only two healthy volunteers had increased values (0.38 and 0.69, according to positive control with 0.35). However, no thrombotic or bleeding events were noted in these two healthy volunteers. The samples with high anti-PF4 Ab were from the ninth and 42nd volunteers (Figure 1B).

Tracing these two volunteers, the platelet counts were 375,000, 347,000, and 358,000/μL for the ninth volunteer, and 250,000, 243,000, and 250,000/μL for the 42nd volunteer at the above three time points, respectively.

### 3.3. Surface Markers of Nonactivated Platelets

Flow cytometry showed that the expression of the MFI of CD42a (glycoprotein IX [GPIX]) decreased 1 day after vaccination compared with the prevaccination values and increased 7 days after injection. However, a significant difference was noted only between 1 day and 7 days after injection (78.29 ± 15.7 vs. 73.12 ± 10.04 vs. 80.32 ± 15.69). Other surface markers including CD41 (glycoprotein IIb/IIIa) and CD42b (glycoprotein Ib) showed no significant differences (Figure 2).

### 3.4. Surface Markers of Activated Platelets

Flow cytometry showed that the percentage of activated GPIIb/IIIa complex (PAC-1) under high ADP or TRAP stimulation increased 1 day after vaccination compared with the prevaccination values (88.17 ± 6.87 vs. 89.67 ± 5.63, *p* = 0.035, under high ADP; 82.79 ± 12.76 vs. 86.42 ± 9.12, *p* = 0.027, under high TRAP). The MFI of PAC-1 under low ADP stimulation increased 7 days after vaccination (57.23 ± 40.05 vs. 67.04 ± 43.27; *p* = 0.022). However, regarding the activated markers of *p*-selectin under low or high TRAP stimulation, the MFI increased 7 days after vaccination (33.49 ± 10.75 vs. 29.80 ± 10.84, *p* = 0.019, under low TRAP; 49.14 ± 15.98 vs. 42.92 ± 17.08, *p* = 0.05, under high TRAP). Other surface markers showed no significant differences (Figure 3).

### 3.5. Leukocyte–Platelet Aggregation

The percentage of neutrophil–platelet and monocyte–platelet aggregations increased 7 days after vaccination (29.11% ± 8.41% vs. 31.44% ± 10.68%, *p* = 0.008, for neutrophil–platelet aggregation; 49.43% ± 14.65% vs. 57.08% ± 18.04%, *p* < 0.001, for monocyte–platelet aggregation); however, the percentage of lymphocyte–platelet aggregation decreased 1 day after vaccination (23.27% ± 5.97% vs. 20.41% ± 6.53% vs. 20.94% ± 6.53%, *p* = 0.019 on day 1 and 0.029 on day 7) (Figure 4).

### 3.6. Cytokine Release

IL-8 and IP-10 significantly increased 1 day after vaccination compared with the prevaccination values (1.44 ± 0.60 vs. 1.80 ± 0.85, *p* = 0.009, for IL-8; 157.98 ± 39.14 vs. 223.19 ± 95.18, *p* < 0.0001, for IP-10). Other cytokines including IFNr, IL-1b, IL-6, IL-15, MCP-1, MIP-1b, RANTES, and TNF-α showed no significant differences between prevaccination and 1 day after vaccination (Figure 5).

## 4. Discussion

Since the initial outbreak in December 2019, SARS-CoV-2 infection and its associated COVID-19 has quickly developed into a pandemic [2,3,4,5]. Elderly patients and those with comorbidities have higher risks of developing critical illness and dying from the disease [6,18]. Thus, vaccination became important in preventing infection or critical illness, and several vaccines were developed. However, various side effects associated with these vaccines have been reported. VITT and vaccine-related ITP were reported after AZ vaccine administration [7,8,9]. The etiologic mechanisms of VITT were suggested to be related to platelet consumption and the progressive thrombogenic state induced by PF4. In this study, activated platelets and inflammation with the release of cytokines played important roles after vaccination, although VITT did not occur. The main finding in this investigation was the occurrence of mild thrombocytopenia with increased platelet activation and inflammation with an elevation in the IL-8 and IP-10 levels and percentage of neutrophil–platelet and monocyte–platelet aggregations after AZ vaccination in healthy volunteers without VITT. The complete study results are presented in Table 2 and Table 3, and the results are also depicted in Figure 2, Figure 3, Figure 4 and Figure 5.

A Norwegian cohort study reported that 1.6% of the 492 subjects experienced mild thrombocytopenia after ChAdOx1 vaccination [19], but the platelet counts were not evaluated at the baseline for comparison. Therefore, whether mild thrombocytopenia occurs after ChAdOx1 vaccination is unclear. One study compared 521 vaccinated Thai participants with 146 nonvaccinated individuals and found that platelet counts decreased significantly after ChAdOx1 vaccination. No significant differences in the levels of anti-PF4 Ab and D-dimer were observed between the vaccinated and nonvaccinated individuals, and no detectable anti-PF4 Abs were found [20]. Similar results were obtained in our study; mild thrombocytopenia occurred 1 day after AZD1222/ChAdOx1 injection. However, we also observed that the condition recovered 7 days after vaccination, which could be attributed to the normal platelet life span of 7–10 days.

Other than thrombocytopenia, platelet activation after vaccination has also been reported in previous studies. Platelet hyper-reactivity has been reported to occur during COVID-19 infections and vaccinations. Several lines of evidence have established a correlation between COVID-19 infection and thrombotic events including thrombocytopenia, platelet hyper-reactivity, and severe bleeding [21,22]. According to Althaus et al., anti-PF4 Abs in patients with VITT significantly increased the procoagulant markers including *p*-selectin and phosphatidylserine externalization compared with those in healthy vaccinated volunteers [23]. Several cases of thrombosis, thrombocytopenia, and severe bleeding have been reported after the administration of the Oxford-AZ vaccine [24]. The WHO and the European Medicines Agency have stated that the association of the vaccine with an increased risk of blood clots is not justifiable and advised continuing the vaccinations [25]. However, in some studies, platelet activation was not observed. According to Youness Limami et al., the Oxford-AZ ChAdOx1 COVID-19 vaccine had no impact on platelet aggregation. Platelet aggregation tests did not indicate enhanced aggregation after vaccination. In response to a suboptimal α-thrombin concentration of 0.05 U/mL, no significant difference in platelet aggregation was recorded [26]. In our study, we used flow cytometry to evaluate the platelet function as it has several advantages such as requiring only a small volume of blood and not requiring centrifugation [27]. To the best of our knowledge, no cohort studies have evaluated platelet surface markers and leukocyte–platelet aggregations. The results of our study revealed that the percentage of PAC-1 expression under high ADP or TRAP stimulation increased 1 day after vaccination. According to a previous study, in inflammatory and thrombotic syndromes, platelets aggregate with circulating leukocytes, particularly monocytes and neutrophils [28]. We also used leukocyte–platelet aggregations to discern the relationship between platelet activation and inflammation. The percentage of neutrophil–platelet and monocyte–platelet aggregations increased 7 days after injection. Accordingly, we found enhanced platelet activation and inflammation after AZD1222/ChAdOx1 injection, particularly 1 week after administration.

The mechanisms and pathogenesis of VITT may be associated with platelet activation, PF4 release, complex formation, and anti-PF4 Ab production. In healthy individuals, PF4 is present in the megakaryocytes and alpha granules of platelets. An early postvaccination event leading to a breach in the tolerance and generation of functionally active anti-PF4 Abs would be necessary to induce platelet activation and the release of PF4. Several studies have focused on the components of the AZ adenovirus vector vaccine that trigger a cascade of early events [29,30]. In a study involving 4000 blood bank donors, the PF4/heparin antibody was evaluated and anti-PF4 Abs was detected at low levels in up to 7% of the donors [31]. Another study reported that approximately 7% of individuals vaccinated against COVID-19 with adenovirus vector-based vaccines or mRNA vaccines had low titers of anti-PF4 Abs; however, these antibodies were not functionally active [32]. In our study, two healthy volunteers had increased levels of anti-PF4 Abs, although no thrombotic or bleeding events were noted.

Regarding inflammation and cytokine release, according to previous studies, COVID-19 infection induces extensive inflammatory responses and increases the secretion of IL-1β, interferon-gamma, IP-10, MCP-1, IL-4, and IL-10. Additionally, intensive care unit (ICU) patients with a severe disease process have higher plasma levels of IL-2, IL-7, IL-10, granulocyte colony-stimulating factor, IP-10, MCP-1, MIP-1a, and TNF-α than non-ICU patients [33,34,35]. The pathogenesis of COVID-19-associated VITT remains incompletely understood. As a trigger of VITT, one critical downstream event may be hyperinflammation induced by the AZ adenovirus vector vaccine [29]. The chemokine IL-8/CXCL8, a potent neutrophil chemoattractant and activator, increases under neutrophil activation and NETosis and is involved in the pathophysiology of VITT [36,37]. According to our results, because platelet activation and inflammation occurred after the administration of the AZD1222/ChAdOx1 vaccine, the release of cytokines is important to understand the mechanism of VITT. In our study, the IL-8 and IP-10 levels increased significantly compared with those before and 1 day after vaccination. IL-8 is a biomarker for the prognosis of patients with COVID-19 [1]. In a study, the serum profiles of 40 cytokines in patients with COVID-19 were evaluated at different disease stages, and IL-8 was found to be better in indicating the progression and status of the disease. Plasma levels of IL-8 were elevated in patients with mild as well as severe COVID-19 and increased with disease progression [1].

According to the aforementioned study, IL-8 plays a key role in COVID-19 infection, and similar results were noted in studies on COVID-19 vaccination. A study revealed increased levels of TNF-α, IL-1b, and IL-8 after the administration of the AZD1222/ChAdOx1 vaccine [14]. We also found that IP-10 increased after AZ vaccine administration. IP-10 is also known as the C–X–C motif chemokine ligand 10 (CXCL10) [38]. The production of TNF-α and CXCL10 in response to viral stimulation has been previously reported, which potentially reflects the polarization of the Th1 phenotype. Increased serum concentrations of TNF-α and CXCL10 have also been observed after the administration of one vaccine dose after a previous infection, which is of note because the serum CXCL10 concentrations are positively correlated with postvaccine antibody responses. Furthermore, IP-10 has been detected after SARS-CoV-2 infection and might underpin vaccine protection against viral variants [39].

The association between IL-8 and activated platelets has been described in a previous study by Zheng-Wei Jian et al. [40]. A total of 120 patients with hypertension were enrolled. Compared with the control group, patients in the hypertension group exhibited higher levels of inflammatory factors including IL-6, IL-8, and TNF-α. The activation of platelets and inflammatory factors is closely related to vascular endothelial function injury in patients with hypertension [40]. According to a previous study on deep vein thrombosis, platelet activation is involved in endothelial cells and serum inflammatory factors, with the regulation of *p*-selectin, GPIIb/IIIa, IL-2, IL-6, and IL-8 [41]. According to a VITT study by Ostrowski et al. [14], specific components of the AZ adenovirus vector may serve as initial triggers of inflammation, platelet activation, and thrombin generation. These may potentially lower the threshold for a cascade of events that trigger complications related to excessive inflammation and platelet and coagulation activation, and also promote the development of VITT when combined with high-titer functionally active PF4 antibodies. According to our results, after the injection of AZD1222/ChAdOx1 vaccines, the AZ adenovirus vector may trigger inflammation and platelet activation by damaging endothelial cells and lowering the threshold for a cascade of events that trigger complications related to excessive inflammation and platelet and coagulation activation, even if VITT does not develop. However, IP-10 was found as the biomarker associated with the severity of COVID-19 disease, not associated with vaccines, and more studies are needed to evaluate the association with platelet activation [14].

We also attempted to analyze the correlation among the side effects, clinical data, platelet activation, and cytokine response. Two groups were compared (i.e., volunteers with minor symptoms and those without any symptoms). The results showed *p*-values > 0.5, and hence, no significant difference.

This study had certain limitations. The research included only a limited number of participants, with only 48 paired samples. All volunteers only received AZD1222/ChAdOx1 vaccines, and comparisons with other COVID-19 vaccines were not performed. Although VITT is mostly noted after AZD1222/ChAdOx1 vaccination, some cases of VITT have also been reported after the administration of other COVID-19 vaccines. Moreover, we only evaluated the cytokine levels 1 day after vaccination. One of the strengths of our study is that it is a cohort study in which prevaccination and 1 and 7 days after vaccination were compared. We uniformly collected data from the same individuals who received only AZ vaccines. The inclusion of a broad range of both plasma- and whole blood-based analyses revealed inflammation and platelet activation, which is also a strength of the study.

## 5. Conclusions

In our study, platelet activation and inflammation with the release of cytokines played important roles after AZD1222/ChAdOx1 vaccination, although no cases of VITT occurred. Mild thrombocytopenia with increased platelet activation with CD42a and PAC-1 expression and increased inflammatory cytokine release with the elevation of IL-8 and IP-10 levels were observed after AZD1222/ChAdOx1 vaccination. Platelet activation and inflammation due to IL-8 and IP-10 are key events after ZD1222/ChAdOx1 vaccination.

## Figures and Tables

**Figure 1 vaccines-11-00456-f001:**
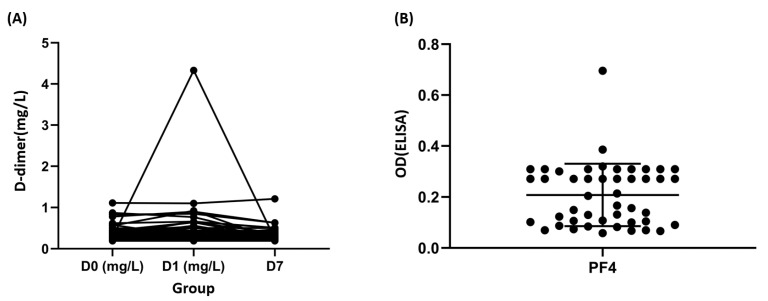
Laboratory characteristics in response to ChAdOx1 nCoV-19 immunization. (**A**) Dynamic changes in the D-dimer levels in donors with ChAdOx1 nCoV-19 vaccination. (**B**) Platelet factor 4 (PF4) optical densities were measured in the plasma of donors. OD: optical density, positivity threshold: OD ≥ 0.4; ELISA: Enzyme-linked immunosorbent assay HAT45G^®®^.

**Figure 2 vaccines-11-00456-f002:**
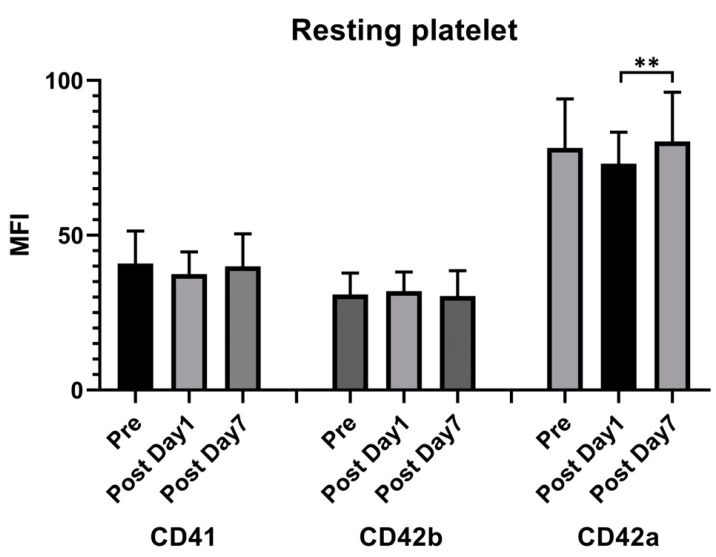
Resting platelet surface receptor expression following the ChAdOx1 nCoV-19 vaccine. Flow cytometry was used to measure the time course of the platelet surface expressions of CD42b, CD42a, and CD41. The presented values are the median fluorescence intensity (MFI) ± standard deviation. ** *p* < 0.01.

**Figure 3 vaccines-11-00456-f003:**
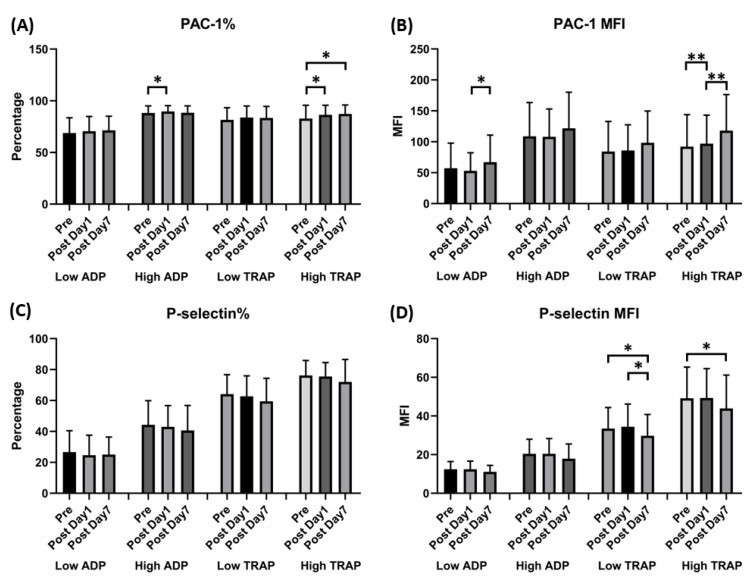
Expression of GPIIb-IIIa (PAC-1) and *p*-selectin in human platelets following ChAdOx1 nCoV-19 vaccination. (**A**) The percentage of platelet surface activated GPIIb-IIIa. (**B**) Platelet surface activated GPIIb-IIIa mean fluorescence intensity (MFI). (**C**) The percentage of platelet surface *p*-selectin. (**D**) Platelet surface *p*-selectin MFI (in the presence of the indicated concentrations of ADP or thrombin receptor-activating peptide [TRAP]). * *p* < 0.05; ** *p* < 0.01.

**Figure 4 vaccines-11-00456-f004:**
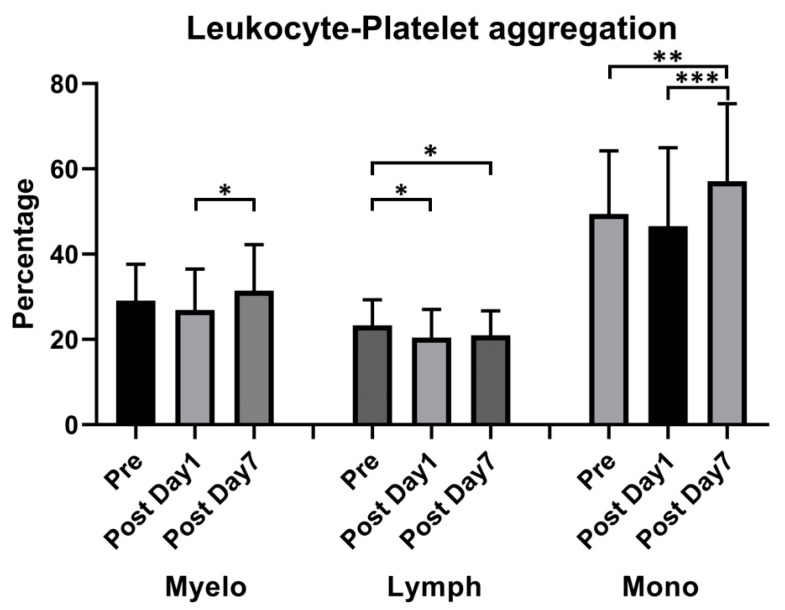
Effect of ChAdOx1 nCoV-19 vaccination on platelet-promoted leukocyte aggregation. Quantitative analysis of platelet–leukocyte aggregates: Myeloid (Myelo), Lymphocyte (Lymph), and Monocyte (mono). * *p* < 0.05; ** *p* < 0.01, *** *p* < 0.001.

**Figure 5 vaccines-11-00456-f005:**
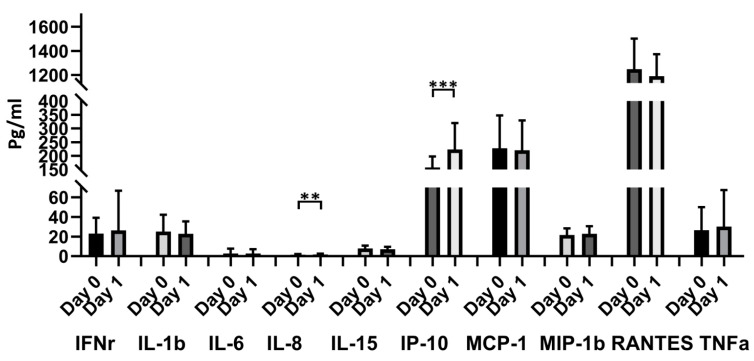
Blood cytokine expressions in the 48 donors with ChAdOx1 nCoV-19 vaccination at different time points. Comparison of 10 cytokines (IFNr, IL-1b, IL-6, IL-8, IL-15, IP-10, MCP-1, MIP-1b, and RANTE) between day 0 and day 1. ** *p* < 0.01, *** *p* < 0.001.

**Table 1 vaccines-11-00456-t001:** The demographic and laboratory characteristics of the 48 donors before and after ChAdOx1 nCoV-19 vaccination.

	Prevaccine	Day 1	Day 7	Pre vs. Day 1	Pre vs. Day 7	Day 1 vs. Day 7
WBC (/μL)	6946 ± 1887 (6401−7492)	6333 ± 1748 (5828−6839)	6653 ± 1654 (6175−7131)	0.007 **	0.141	0.072
Hb (g/dL)	12.99 ± 1.32 (12.61−13.37)	13.03 ± 1.32 (12.65−13.41)	13.09 ± 1.33 (12.70−13.47)	0.245	0.050	0.418
PLT (×10^3^/μL)	285 ± 65 (266−303)	276 ± 66 (257−295)	286 ± 64 (267−304)	0.018 *	0.705	0.039 *
MPV (fL)	12.24 ± 1.23 (9.88−10.60)	10.13 ± 1.14 (9.80−10.46)	9.92 ± 1.03 (9.63−10.22)	0.205	0.002 **	0.016 *
PDW	10.69 ± 1.70 (10.20−11.18)	10.87 ± 1.52 (10.43−11.31)	10.64 ± 1.52 (10.21−11.08)	0.407	0.415	0.064
Myeloid (%)	60.8 ± 7.83 (58.5− 63.1)	59.5 ± 7.80 (57.2−61.7)	59.9 ± 7.70 (57.7−62.2)	0.206	0.344	0.666
Lymphocyte (%)	30.6 ± 7.15 (28.5−32.7)	31.8 ± 7.10 (29.7−33.8)	32.0 ± 7.14 (29.9−34.0)	0.207	0.110	0.834
Monocyte (%)	5.54 ± 1.31 (5.16−5.92)	5.95 ± 1.43 (5.53−6.36)	5.16 ± 1.10 (4.84−5.48)	0.031 *	0.019 *	<0.001 **

Mean ± SE (95% CI); * *p* < 0.05; ** *p* < 0.01. WBC: white blood cell count; Hb: hemoglobin; PLT: platelet; MPV: mean platelet volume; PDW: platelet distribution width.

**Table 2 vaccines-11-00456-t002:** Total study results including D-dimer, PF4, MFI of resting platelets, leukocyte–platelet aggregation, expression of GPIIbIIIa (PAC-1), and *p*-selectin in human platelets before and following the ChAdOx1 nCoV-19 vaccine.

				*p*-Value		
	Day 0	Day 1	Day 7	Day 0 vs. Day 1	Day 0 vs. Day 7	Day 1 vs. Day 7
D-dimer (mg/dL)	0.41 ± 0.19	0.52 ± 0.62	0.35 ± 0.18	-	-	-
PF4 (OD)	-	-	0.21 ± 0.12	-	-	-
**Resting Platelet**						
CD41 MFI	40.90 ± 10.39	37.48 ± 7.10	40.99 ± 10.33	0.052	0.295	0.088
CD42b MFI	30.91 ± 6.84	31.98 ± 6.04	30.35 ± 8.08	0.153	0.602	0.283
CD42a MFI	78.29 ± 15.70	73.12 ± 10.04	80.32 ± 15.69	0.093	0.189	0.007 **
**Leukocyte–Platelet Aggregation (Percentage %)**						
Myeloid%	29.11 ± 8.41	26.91 ± 9.45	31.44 ± 10.68	0.128	0.078	0.008 **
Lymphocyte%	23.27 ± 5.97	20.41 ± 6.53	20.94 ± 5.70	0.019 *	0.029 *	0.605
Monocyte%	49.43 ± 14.65	46.54 ± 18.21	57.08 ± 18.04	0.396	0.009 **	<0.001 **
**Activated Platelets**						
CD42b MFI	77.20 ± 17.06	83.72 ± 18.44	80.63 ± 27.94	0.042 *	0.602	0.629
FSC	101.28 ± 20.51	98.21 ± 21.19	108.12 ± 25.42	0.667	0.004 **	<0.001 **
**PAC-1 Percentage (%)**						
Low ADP	68.85 ± 14.61	70.55 ± 14.16	71.46 ± 13.41	0.258	0.223	0.7839
High ADP	88.17 ± 6.87	89.67 ± 5.63	88.47 ± 6.58	0.035 *	0.785	0.1898
Low TRAP	81.60 ± 11.52	83.80 ± 11.00	83.46 ± 10.92	0.053	0.273	0.8777
High TRAP	82.79 ± 12.76	86.42 ± 9.12	87.37 ± 8.43	0.027 *	0.033 *	0.5827
**PAC-1 MFI**						
Low ADP	57.23 ± 40.05	53.04 ± 28.87	67.04 ± 43.27	0.433	0.123	0.022 *
High ADP	108.94 ± 53.79	108.05 ± 44.44	121.78 ± 57.65	0.313	0.125	0.156
Low TRAP	84.33 ± 48.02	85.97 ± 40.87	98.46 ± 50.63	0.608	0.057	0.058
High TRAP	92.21 ± 51.15	97.03 ± 45.39	117.98 ± 57.78	0.263	0.004 **	0.004 **
***p*-Selectin Percentage (%)**						
Low ADP	26.60 ± 13.69	24.66 ± 12.71	25.03 ± 11.21	0.377	0.506	0.854
High ADP	44.32 ± 15.38	42.99 ± 13.53	40.65 ± 16.01	0.584	0.236	0.584
Low TRAP	64.11 ± 12.49	62.72 ± 13.12	59.42 ± 14.77	0.557	0.069	0.286
High TRAP	76.24 ± 9.54	75.51 ± 8.90	72.01 ± 14.37	0.654	0.072	0.121
***p*-Selectin MFI**						
Low ADP	12.40 ± 3.96	12.40 ± 4.20	11.12 ± 3.31	0.867	0.061	0.092
High ADP	20.42 ± 7.49	20.43 ± 7.78	17.91 ± 7.49	0.925	0.062	0.153
Low TRAP	33.49 ± 10.75	34.44 ± 11.59	29.80 ± 10.84	0.608	0.019 *	0.049 *
High TRAP	49.14 ± 15.98	49.29 ± 15.04	42.92 ± 17.08	0.974	0.050 *	0.139

Mean ± SE (95% CI); * *p* < 0.05; ** *p* < 0.01 MFI: mean fluorescence intensity; PAC-1: GPIIbIIIa; ADP: adenosine diphosphate; TRAP: thrombin receptor-activating peptide.

**Table 3 vaccines-11-00456-t003:** Blood cytokine expression in the 48 donors with the ChAdOx1 nCoV-19 vaccine at different time points.

	Day 0	Day 1	*p*-Value
IFNr	23.90 ± 15.89	26.39 ± 39.92	0.576
IL-1b	25.12 ± 16.92	22.86 ± 12.44	0.219
IL-6	2.67 ± 4.94	2.69 ± 4.33	0.972
IL-8	1.44 ± 0.60	1.80 ± 0.85	0.009 **
IL-15	7.81 ± 3.00	7.06 ± 2.34	0.109
IP-10	157.98 ± 39.14	223.19 ± 95.18	<0.001 **
MCP-1	228.05 ± 118.35	220.27 ± 108.03	0.353
M1P-1b	21.51 ± 6.74	22.89 ± 7.67	0.147
RANTES	1248.6 ± 249.59	1190.4 ± 179.33	0.192
TNFa	26.60 ± 23.09	30.10 ± 36.93	0.214

Mean ± SE (95% CI); ** *p* < 0.01.

## Data Availability

The data analyzed in this study are subject to the following licenses/restrictions: Due to the IRB, raw data were generated at Hualien Tzu Chi Hospital. Derived data supporting the findings of this study are available from the corresponding author Yi-Feng Wu on request. Requests to access these datasets should be directed to Yi-Feng Wu, wuyifeng43@gmail.com.

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
