# Peer review of "Platelet Activation and Cytokine Release of Interleukin-8 and Interferon-Gamma-Induced Protein 10 after ChAdOx1 nCoV-19 Coronavirus Vaccine Injection"

_vaccines, 2023, doi:10.3390/vaccines11020456_

Round 1

Reviewer 1 Report

This manuscript reported by Wu and co-workers is to investigate the impact of AZD1222/ChAdOx1 (AZ) vaccination on inflammation and platelet activation. The main targets evaluated in this manuscript included antiplatelet factor IV antibody, platelet activation and aggregation, cytokine release, etc. There were 48 individuals who were recruited in Taiwan as participants without Vaccine-induced Immune thrombocytopenia and thrombosis (VITT). Their blood samples were studied before vaccination and also 1 and 7 days after AZ vaccination. The main findings were that leukocyte and platelet counts decreased after 1-day vaccination and then recovered after 7 days. AZ vaccination induced platelet activation and enhanced inflammation associated with an increase of PAC-1 expression and elevation of Interleukin 8 and interferon-gamma-induced protein 10. 

Though this manuscript provides many statistical data, the discussion is not profound and clear enough. It is suggested to add a table in the Discussion to list all the numbers resulting from Figures 2, 3, 4, and 5. In addition, some minor points indicated below need to be revised and improved.

1.     The full names of VITT and PAC-1 (p. 1, lines 18 and 21) in the abstract should be provided for their first use.

2.     The citation sources and their associated time regarding to the numbers of incidences and fatalities caused by COVID-19 should be specified in the sentences on p. 1, lines 34 and 35.

3.     The baseline characteristics of the participants, such as gender, age and their mean age with standard deviation, have to be provided in the text. Those data are fundamental information for the clinical studies.

4.     As the symptoms of the participants have been recorded after vaccination (p. 2, lines 76 and 77), this information is suggested to be added in the text as references and to be discussed if there is a correlation between the severity of symptoms and biological data.

All of requested revisions should be done before its acceptance by Vaccines.

Author Response

We thank the reviewer for reminding us of this important issue. For the English editing, the revised manuscript had been under English revisions. (Editing Certificate as attached)

We tried to explain clearly in the discussion section, and added more discussion between vaccination, platelets, and cytokines. We added Table 2 and Table 3 for total study results in the Discussion, and the results also showed as Figures 2, 3, 4, and 5.

Table 2. Total study results, including D-dimer, PF4, MFI of resting platelets, leukocyte-platelet aggregation, expression of GPIIbIIIa (PAC-1), and P-selectin in human platelets before and following the ChAdOx1 nCoV-19 vaccine.

Table 3. Blood cytokine expression in the 48 donors with ChAdOx1 nCoV-19 vaccine at different time points.

The discussion section was added with “The association between IL-8 and activated platelets have been described in a previous study by Zheng-Wei Jian et al. A total of 120 patients with hypertension were enrolled. Compared with the control group, patients in the hypertension group exhibited higher levels of inflammatory factors, including IL-6, IL-8, and TNF-α. The activation of platelets and inflammatory factors is closely related to vascular endothelial function injury in patients with hypertension. According to a previous study on deep vein thrombosis, platelet activation is involved in endothelial cells and serum inflammatory factors, with the regulation of P-selectin, GPIIb/IIIa, IL-2, IL-6, and IL-8. According to a VITT study by Ostrowski et al., specific components of the AZ adenovirus vector may serve as initial triggers of inflammation, platelet activation, and thrombin generation. These may potentially lower the threshold for a cascade of events that trigger complications related to excessive inflammation and platelet and coagulation activation and also promote the development of VITT when combined with high-titer functionally active PF4 antibodies. According to our results, after the injection of AZD1222/ChAdOx1 vaccines, AZ adenovirus vector may trigger inflammation and platelet activation by damaging endothelial cells and lowering the threshold for a cascade of events that trigger complications related to excessive inflammation and platelet and coagulation activation even if VITT does not develop. However, IP-10 was found as the biomarker associated with the severity of COVID-19 disease, not associated with vaccines, and more studies are needed to evaluate the association with platelet activation.”

1. We changed to “Vaccine-induced immune thrombotic thrombocytopenia (VITT)” at Line 18 and “activated GPIIb/IIIa complex (PAC-1)” at Line 21~22.

2. We added detailed information with “Tracing the Weekly COVID-19 Epidemiological Update until November 08, 2022, revealed that over 630 million individuals were confirmed to have the disease and that more than 6.6 million associated deaths were reported from all six World Health Organization (WHO) regions.”

3. We added the characteristics in section 3. Results / 3.1 Clinical data as “The baseline characteristics of the volunteers included 13 men and 35 women. Their mean age was 39.1 ± 9.8 years.”

4. We also added the characteristics in section 3. Results / 3.1 Clinical data “The side effects included 50% local reddish and pain over injection sites, 33.30% fatigue, 12.50% headache, 25% muscle pain, 16.70% chills, 12.50% joint pain, and 8.30% fever. The above minor side effects were noted in 72.9% (N=35) of the volunteers. No severe side effects, including severe allergy, drowsiness, abdominal pain, poor appetite, lymphadenopathy, or VITT, were found.”

And to our data, there was no correlation between the severity of symptoms and biological data. In the section of Discussion, we also added as “We also attempted to analyze the correlation among side effects, clinical data, platelet activation, and cytokine response. Two groups were compared, i.e., volunteers with minor symptoms and those without any symptoms. The results showed p-values > 0.5, and hence, no significant difference.”

Reviewer 2 Report

The paper demonstrated the platelet activation and cytokine release after ChAdOx1 nCoV-19 coronavirus vaccination in healthy individuals. This study is highly relevant to understand vaccine-induced immune thrombocytopenia thus, the audience of the journal will have great interest.

1) Describe the participant characteristics, such as sex and body mass index if available. The platelet activation and cytokine release might be influenced by the participant's characteristics. In addition, please indicate whether the ChAdOx1 nCoV-19 coronavirus vaccine was used for the primary series of COVID vaccination or mixed procedure with the mRNA vaccine.

2) Laboratory data were repeatedly measured so that please make sure to implement the repeated-measure one-way ANOVA in this study for statistical evaluation, not ordinary one-way ANOVA (Line 135).

3) A donor with elevated D-dimer (Line 155) and two with anti-PF4 antibody were observed. It will be appreciated if the alternation of platelet counts in the three donors is shown.

4) The changes in the platelet count, IL-8 and IL-10 were mild but showed statistical significance in this study cohort. Discuss the possibility of subject variation on the platelet activation and cytokine response.

5) The last sentence in the conclusion (Line 347–348) should be revised to be specific to the study results.

Author Response

We thank the reviewer for reminding us of this important issue. For the English editing, the revised manuscript had been under English revisions. (Editing Certificate as attached)

1. We added the characteristics in section 3. Results / 3.1 Clinical data as “The baseline characteristics of the volunteers included 13 men and 35 women. Their mean age was 39.1 ± 9.8 years.” But we did not record the body mass index for this study.

And in section 2. Materials and Methods / 2.1 Healthy volunteers, we added “None of the volunteers received other kinds of COVID-19 vaccines, including mRNA vaccines.”

2. Thanks to the reviewer’s suggestion, we changed to “Laboratory data, platelet surface antigens, platelet–leukocyte aggregations, and cytokines between the different time points were compared using one-way ANOVA” in the “2.7 Statistical analysis”.

3. For the detailed data, we added “Tracing this volunteer, the platelet counts were 206000, 193000, and 223000/μL at the above three time points” for the 7th volunteer with the elevation D-dimer. And “Tracing these two volunteers, the platelet counts were 375000, 347000, and 358000/μL for the 9th volunteer, and 250000, 243000, and 250000/μL for the 42nd volunteer at the above three time points” for the volunteers of elevation of anti-PF4.

4. We thank the reviewer for reminding us of this important issue. The discussion section was added with “The association between IL-8 and activated platelets have been described in a previous study by Zheng-Wei Jian et al. A total of 120 patients with hypertension were enrolled. Compared with the control group, patients in the hypertension group exhibited higher levels of inflammatory factors, including IL-6, IL-8, and TNF-α. The activation of platelets and inflammatory factors is closely related to vascular endothelial function injury in patients with hypertension. According to a previous study on deep vein thrombosis, platelet activation is involved in endothelial cells and serum inflammatory factors, with the regulation of P-selectin, GPIIb/IIIa, IL-2, IL-6, and IL-8. According to a VITT study by Ostrowski et al., specific components of the AZ adenovirus vector may serve as initial triggers of inflammation, platelet activation, and thrombin generation. These may potentially lower the threshold for a cascade of events that trigger complications related to excessive inflammation and platelet and coagulation activation and also promote the development of VITT when combined with high-titer functionally active PF4 antibodies. According to our results, after the injection of AZD1222/ChAdOx1 vaccines, AZ adenovirus vector may trigger inflammation and platelet activation by damaging endothelial cells and lowering the threshold for a cascade of events that trigger complications related to excessive inflammation and platelet and coagulation activation even if VITT does not develop. However, IP-10 was found as the biomarker associated with the severity of COVID-19 disease, not associated with vaccines, and more studies are needed to evaluate the association with platelet activation.”

5. As the suggestion of reviewers, the last sentence was changed to “Platelet activation and inflammation due to IL-8 and IP-10 are key events after ZD1222/ChAdOx1 vaccination.”

Round 2

Reviewer 1 Report

The amendments and explanations provided by Wu and co-workers in this revised manuscript are clear. Two tables (i.e., Tables 2 and 3) are added to provide the data related to the platelet activation and cytokine release.  These results are also incorporated into the Discussion section and some further discussion is made therein.

Overall, authors have answered the questions raised by the reviewer point-by-point. However, there are three minor points, which need to be revised as indicated below.

1. p.4, lines 147-148: 50% should be revised to 50.0%; 33.30% should be 33.3%; 12.50% should be 12.5%; 25% should be 25.0%; 16.70% should be 16.7%; 12.50% should be 12.5%; 8.30% which is correct because of significant figures.

2. p. 8, Table 2, “Resting Platelet” is suggested to be revised to “Resting platelet” and “Leukocyte-Platelet aggregation” is revised to “Leukocyte-platelet aggregation” to let the subtitles be consistent.

3. p. 11, lines 355-356, add reference(s) for the sentence “…. a previous study by Zheng-Wei Jian et al.”; line 360, add reference(s) for “According to a previous study ….”; lines 362-363, add reference(s) for “… a VITT study by Ostrowski et al.”

After all the revision is carried out, the revised version is recommended to be accepted for publication by Vaccines.

Author Response

Thanks for the reviewer’s suggestion. We corrected the points as below.

1. On p.4, the manuscript was changed to “The side effects included 50.0% local reddish and pain over injection sites, 33.3% fatigue, 12.5% headache, 25.0% muscle pain, 16.7% chills, 12.5% joint pain, and 8.30% fever.”

2. On p. 8, Table 2 was changed to “Resting platelet” and “Leukocyte-platelet aggregation”.

3. On p. 11 and reference, we added three new references:

[40] Zheng-Wei Jian, Xiao-Ming Zhang, Guan-Shen Huang. Clinical value of the platelet and inflammatory factor activation in vascular endothelial injury in essential hypertension. Clin Hemorheol Microcirc. 2022 Nov 26. doi: 10.3233/CH-221638.

[41] Jianhui Wu, Haimei Zhu, Guodong Yang, Yuji Wang, Yaonan Wang, Shurui Zhao, et al. IQCA-TAVV: To explore the effect of P-selectin, GPIIb/IIIa, IL-2, IL-6 and IL-8 on deep venous thrombosis. Oncotarget. 2017 Aug 24;8(53):91391-91401.

[42] Sisse R Ostrowski, Ole S Søgaard, Martin Tolstrup, Nina B Stærke, Jens Lundgren, Lars Østergaard, et al. Inflammation and Platelet Activation After COVID-19 Vaccines - Possible Mechanisms Behind Vaccine-Induced Immune Thrombocytopenia and Thrombosis. Front Immunol. 2021 Nov 23;12:779453.

We thank the editor and reviewers for the extensive assessment of the manuscript, and the important and helpful comments and suggestions.